**Subject Area:**
molecular biology/genetics

marine fish, population structure, local adaptation, parallel evolution

**Authors for correspondence:**
Tianxiang Gao
e-mail: gaotianxiang0611@163.com
Xiumei Zhang
e-mail: xiumei1227@163.com

# Population genomics reveals possible genetic evidence for parallel evolution of *Sebastiscus marmoratus* in the northwestern Pacific Ocean

Shengyong Xu[1], Takashi Yanagimoto[2], Na Song[3], Shanshan Cai[1], Tianxiang Gao[1] and Xiumei Zhang[1]

[1]National Engineering Research Center For Marine Aquaculture, Zhejiang Ocean University, 1st Haidanan Road, Zhoushan, Zhejiang, People's Republic of China
[2]National Research Institute of Fisheries Science, 2-12-4, Fukuura, Kanazawa, Yokohama, Japan
[3]Institute of Evolution and Marine Biodiversity, Ocean University of China, 5th Yushan Road, Qingdao, Shandong, People's Republic of China

SX, 0000-0001-8475-7315; TG, 0000-0001-8797-4147

Understanding patterns of population diversity and structuring among marine populations is of great importance for evolutionary biology, and can also directly inform fisheries management and conservation. In this study, genotyping-by-sequencing was used to assess population genetic diversity and connectivity of *Sebastiscus marmoratus*. Based on 130 individuals sampled from 10 locations in the northwestern Pacific Ocean, we identified and genotyped 17 653 single-nucleotide polymorphisms. The patterns of genetic diversity and population differentiation suggested that the Okinawa Trough might be the ancestral centre of *S. marmoratus* after the Last Glacial Maximum. A shallow genetic structure was observed among sampled populations based on the implemented structuring approaches. Surprisingly, we detected genetic homogeneity in two population pairs (i.e. Xiamen–Niigata and Zhuhai–Iki Island), in which populations have large geographical and latitudinal intervals. Population structure and allele frequency distribution based on outlier loci also mirrored the observed genetic homogeneity in the above-mentioned population pairs. Integrated with biological, environmental and genomic data, our results provide possible genetic evidence for parallel evolution. Our study also provides new perspectives on the population structure of *S. marmoratus*, which could have important implications for sound management and conservation of this fishery species.

## 1. Introduction

Determining parallel evolution is important because it can reveal the nature of the ecological and evolutionary forces that shape biodiversity [1]. Genetic parallelism appears to be ubiquitous and occurs at all taxonomic levels [2–4]. When species adapt to local biotic and abiotic conditions, similar selective pressures will lead to identical or similar adaptive changes in distantly related species or populations [5,6].

Parallel evolution provides valuable systems for studying the genetics of new adaptations [3], and identification of the genetic basis for parallel adaptation is a prominent goal in evolutionary biology [7]. Recent technological advances in population-scale high-throughput sequencing provide powerful tools to explore parallel evolution at genomic scales [8]. In the marine realms, applying population genomics approaches, parallel evolutionary processes have been recently observed in threespine stickleback *Gasterosteus aculeatus* [8–10], European anchovy *Engraulis encrasicolus* [11], Atlantic cod *Gadus morhua* [12], steelhead/rainbow trout *Oncorhynchus mykiss* [7] and Atlantic herring *Clupea harengus* [13]. However, such evolutionary parallelism studies are generally restricted to similar

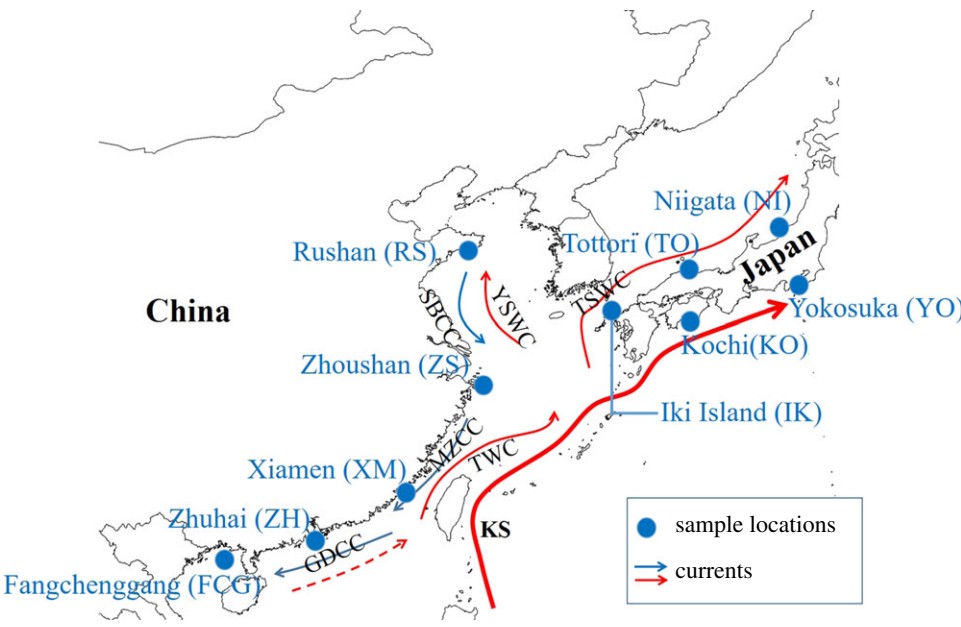

**Figure 1.** Schematic map showing sample locations of *S. marmoratus* and contemporary main currents of the northwestern Pacific Ocean in winter. KS, Kuroshio Current; TWC, Taiwan Warm Current; TSWC, Tsushima Warm Current; YSWC, Yellow Sea Warm Current; SBCC, Subei Coastal Current; MZCC, Minzhe Coastal Current; GDCC, Guangdong Coastal Current. The map was generated using ArcGIS 10.2, made with Natural Earth. The coastline data are available on the Internet at http://www.naturalearthdata.com/downloads/. The main currents follow the description in Liu [14].

latitudinal gradients on both sides of the Atlantic Ocean. Therefore, we wanted to investigate whether parallel evolution could be detected within large latitudinal scales.

In the northwestern Pacific Ocean (NWP), with the influence of the Kuroshio Current System (KCS), the ecological conditions between southeastern Chinese (SEC) coasts and Japanese (JPN) coasts are roughly homogeneous [14]. It is thought that parallel evolution could be possibly detected in marine organisms between the two regions. The marbled rockfish *Sebastiscus marmoratus* is probably an ideal candidate for the detection of parallel evolution. With low genetic background noise, the genetic homogeneity in this species [15] allows us to identify selective outliers associated with local adaptation [13]. Reported biological studies have demonstrated that the mating seasons of *S. marmoratus* are roughly identical between SEC and JPN regions [16,17]. It is commonly known that mating seasons largely influence the reproduction and population recruitment of ovoviviparous fish, rather than spawning seasons. The shared reproductive features could be an indicator of environmental and genetic similarity. In addition, our recent population genetic study explicated the close relationship between sampled individuals from Xiamen and Japan [18]. Therefore, in the present study, with the objective of delineating a precise genetic pattern of *S. marmoratus* populations, we applied a genotyping-by-sequencing (GBS) approach to investigate the population diversity and structuring of 10 populations (five Chinese populations and five Japanese populations) sampled from the NWP. The inclusion of *S. marmoratus* samples from SEC and JPN regions may provide possible evidence for parallel evolution of this species.

# 2. Material and methods

## 2.1. Sample collection

Adult *S. marmoratus* were obtained from 10 locations along the NWP (figure 1 and table 1). The samples were collected by trawl net or hook fishing in offshore waters, thus ensuring that the samples collected were representative of the local populations. Muscle tissues were preserved in 95% ethanol. All samples were collected in accordance with national legislation.

## 2.2. DNA extraction, GBS library preparation and sequencing

DNA isolation was accomplished by a standard phenol–chloroform extraction protocol, followed by RNase A treatment. The GBS libraries were constructed following Elshire *et al.* [19]. Briefly, the DNA was digested with both high-fidelity *NlaIII* and *MseI* restriction enzymes. A total of 10 libraries were created by uniquely barcoding each of the individuals from the respective site and then pooling these individually barcoded samples. The barcodes used were six nucleotides in length. The libraries were pooled for multiplexed polymerase chain reactions (PCRs), and then the PCR products were purified. The sequencing was performed in three lanes of an Illumina HiSeq2500 platform, using 150-bp paired-end reads. The library preparation and sequencing processes were performed commercially at Novogene Co. Ltd in Beijing, China.

## 2.3. SNP calling and filtering

Raw sequences were parsed, trimmed and demultiplexed using the bioinformatics tool Trimmomatic 0.36 [20] with default parameters. A draft genome sequence of *S. marmoratus* assembled using SOAPdenovo2 software [21] based on whole-genome resequencing data was used as the reference. All quality-filtered reads were sorted and aligned to the reference sequence using the bwa-mem algorithm in BWA 0.7.12 [22] with default parameters. After the alignment, single-nucleotide polymorphism (SNP) calling was performed using SAMtools 1.3.1 [23]. SNP filtering was produced using VCFtools [24] with the following parameters. Filtering criteria: (i) the SNP was called in 90% of individuals, (ii) the minor

**Table 1.** Sample information of *S. marmoratus* in the northwestern Pacific Ocean.

| sample site | code | country | coordinates | sample size | collection date |
|---|---|---|---|---|---|
| Rushan, Yellow Sea | RS | China | 36°43′ N, 121°39′ E | 20 | Oct 2015 |
| Zhoushan, East China Sea | ZS | China | 30°03′ N, 122°21′ E | 20 | Nov 2015 |
| Xiamen, East China Sea | XM | China | 24°22′ N, 118°13′ E | 10 | Mar 2016 |
| Zhuhai, South China Sea | ZH | China | 22°16′ N, 113°36′ E | 10 | Sep 2015 |
| Fangchenggang, South China Sea | FCG | China | 21°30′ N, 108°21′ E | 20 | Oct 2015 |
| Niigata, Sea of Japan | NI | Japan | 37°57′ N, 138°54′ E | 10 | Jul 2015 |
| Kochi, Pacific Ocean | KO | Japan | 33°27′ N, 133°34′ E | 10 | Sep 2012 |
| Yokosuka, Pacific Ocean | YO | Japan | 35°17′ N, 139°43′ E | 10 | Nov 2011 |
| Tottori, Sea of Japan | TO | Japan | 35°33′ N, 134°12′ E | 10 | Jun 2015 |
| Iki Island, Sea of Japan | IK | Japan | 33°47′ N, 129°38′ E | 10 | Sep 2012 |

allele frequency (MAF) was greater than 5%, (iii) only two alleles were present, (iv) sites that contained an indel were excluded, and (v) sites that failed the Hardy–Weinberg equilibrium (HWE) test at $p < 0.05$ were excluded. The parameter scripts of BWA, SAMtools and VCFtools are shown in the electronic supplementary material. We also tested for departures from linkage disequilibrium (LD) expectations within each sample site and chose to exclude loci that exhibited strong LD ($r^2 > 0.8$ [25]). Testing for LD made use of the TASSEL 5.0 software [26]. All datasets were reformatted using PGDSpider 2.0.5.2 [27].

## 2.4. Outlier detection

To identify outlier loci putatively under selection, two different approaches were used. Firstly, we used the *fdist* approach implemented in Lositan [28]. The probability of each locus $F_{st}$ belonging to the neutral distribution is used to classify loci into one of three selection categories: neutral selection (0.1–0.9), balancing selection (less than 0.1) and divergent selection (greater than 0.995) [29]. Outlier analyses were based on 60 000 simulations assuming an infinite allele mutation model and using neutral mean $F_{st}$, 0.95 confidence intervals and a false discovery rate of 0.1. Five independent runs were performed to further reduce false positives. Secondly, we used a Bayesian model-based approach in BayeScan 2.1 [30], which has been shown to have lower type 1 error rates [31]. Analyses were conducted using default settings. Loci under selection were defined as those with a false discovery rate (FDR) of 0.1. Loci both under divergent selection in Lositan and with an FDR of 0.1 in BayeScan were considered as outliers. Following removal of these outliers and those under balancing selection, the retained loci were set as the non-outlier dataset. The non-outlier dataset was used in population genetic analyses.

## 2.5. Population genetic analyses

The program Arlequin 3.5 [32] was used to estimate average SNP diversity ($\pi$) and expected heterozygosity ($H_e$) of each population. Given that genetic diversity indexes are normally correlated to sample size [33], rarefaction of sample size (10 individuals per population) was randomly subsampled and the genetic diversity indexes were recalculated based on the non-outlier dataset. Allele frequencies of outliers across all populations were calculated using Arlequin and then the differences were compared SNP by SNP between populations. Hierarchical clustering using Euclidean distance with the Ward clustering method was performed on allele frequencies using the R package *pheatmap* to detect population homogeneity based on allele frequencies. The pairwise fixation index ($F_{st}$) for each of the two populations was estimated using Arlequin software and the significance of each pairwise $F_{st}$ value was assessed by 10 000 bootstrap permutations [32]. To correct for multiple hypothesis testing we applied a Benjamini–Hochberg FDR ($\alpha = 0.01$) correction [34] using the R function *p.adjust* in the package *stats*.

Population structure was investigated by using the program Admixture 1.3.0 [35]. The best value of the coancestry cluster ($K$) was estimated using a cross-validation procedure in the Admixture software. The best value of the coancestry cluster exhibited the lowest cross-validation error (CVE) compared with the other cluster values. In order to further estimate the substructure of *S. marmoratus* populations, we increased the coancestry clusters spanning from 2 to 10 and ran the analysis with 10 000 iterations. Principal component analysis (PCA) was also implemented using the R package *adegenet* [36] to determine whether sampled individuals reflected a history of differentiated populations. Finally, a population-based neighbour-joining (NJ) tree was constructed using the program Populations [37] based on estimates of genetic distance among populations (Cavalli-Sforza and Edwards chord distance, $D_c$) with 1000 bootstrap replications on individuals and visually displayed using SplitsTree 4.14.4 [38].

## 2.6. Gene prediction and functional annotation

The contigs containing outlier loci were used as queries in nucleotide searches with BlastX ($E$-value $1 \times 10^{-3}$) against the NR database at the National Center for Biotechnology Information (NCBI) website. In the case of multiple hits, the best match was chosen. Then, the functional annotations of these genes were obtained using Blast2GO [39]. This software conducts Blast similarity searches and maps Gene Ontology (GO) for homologous sequences. Blast2GO produces GO annotations as well as corresponding enzyme commission numbers (EC) for sequences with $E$-values $< 1 \times 10^{-6}$, annotation cut-off values greater than 55 and a GO weight greater than 5.

royalsocietypublishing.org/journal/rsob Open Biol. **9**: 190028

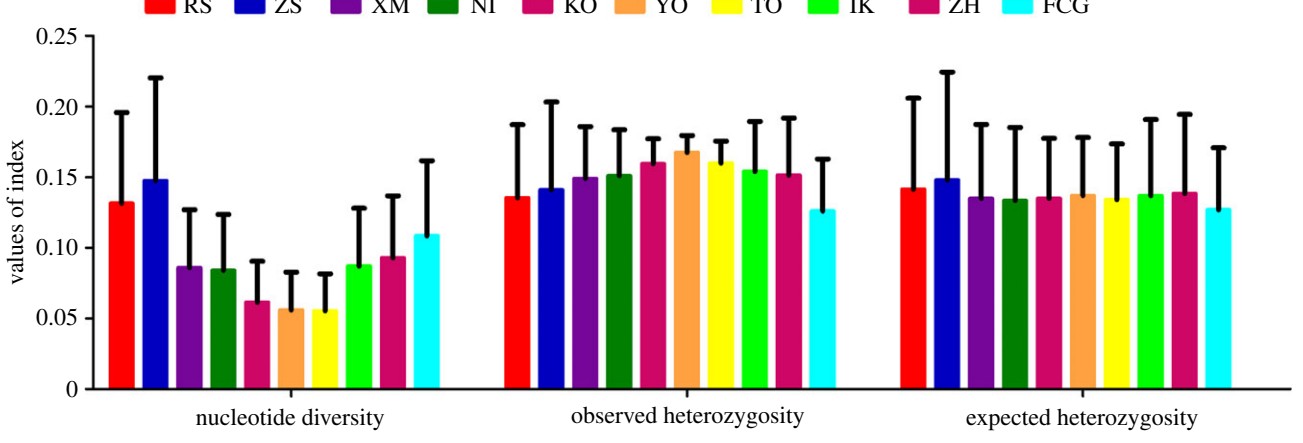

**Figure 2.** Genetic diversity index among *S. marmoratus* populations based on the non-outlier dataset.

**Table 2.** Pairwise $F_{st}$ estimates among populations using non-outlier loci. Significant values are shown in italics ($p < 0.05$, FDR $\alpha = 0.01$).

| | NI | KO | YO | TO | IK | RS | ZS | XM | ZH | FCG |
|---|---|---|---|---|---|---|---|---|---|---|
| NI | — | | | | | | | | | |
| KO | 0.0107 | — | | | | | | | | |
| YO | 0.0121 | −0.0055 | — | | | | | | | |
| TO | 0.0088 | 0.0032 | 0.0028 | — | | | | | | |
| IK | 0.0074 | 0.0103 | 0.0126 | 0.0115 | — | | | | | |
| RS | 0.0139 | 0.0225 | 0.0255 | 0.0257 | 0.0113 | — | | | | |
| ZS | 0.0120 | 0.0292 | 0.0319 | 0.0315 | 0.0172 | 0.0105 | — | | | |
| XM | −0.0010 | 0.0122 | 0.0150 | 0.0118 | 0.0081 | 0.0138 | 0.0112 | — | | |
| ZH | 0.0092 | 0.0114 | 0.0132 | 0.0129 | 0.0047 | 0.0095 | 0.0149 | 0.0089 | — | |
| FCG | 0.0052 | 0.0155 | 0.0175 | 0.0142 | 0.0112 | 0.0162 | 0.0147 | 0.0052 | 0.0079 | — |

# 3. Results

The 130 individual *S. marmoratus* in this study (figure 1 and table 1) were sampled from Rushan (RS, $n = 20$), Zhoushan (ZS, $n = 20$), Xiamen (XM, $n = 10$), Zhuhai (ZH, $n = 10$), Fangchenggang (FCG, $n = 20$), Niigata (NI, $n = 10$), Kochi (KO, $n = 10$), Yokosuka (YO, $n = 10$), Tottori (TO, $n = 10$) and Iki Island (IK, $n = 10$).

## 3.1. Genotyping, SNP calling and filtering

The GBS sequencing produced 212.8 million high-quality reads, with a mean of 1.64 million reads per individual (electronic supplementary material, table S1). Quality-filtered reads were mapped against the assembled genome. A total of 563 880 putative SNPs were produced among our samples and 17 653 SNPs were retained following filtering steps. Of these loci, 1.0% ($n = 180$) were identified as outliers, 0.8% ($n = 141$) were under balancing selection, and the rest ($n = 17 332$) were considered as the non-outlier dataset.

## 3.2. Gene annotation analyses

Genome-wide scans for selection identified 180 outlier loci. The BlastX analysis showed 14 contigs containing outliers corresponding to known proteins in the NR database and eight contigs were functionally annotated. The low proportion of annotated loci might be due to the combination of a lack of a high-quality reference genome and short assembled contigs of GBS reads. The annotated loci had homology to proteins associated with metabolic activities such as transferase activities, protein binding, motor activities and oxidoreductase activities, among others (electronic supplementary material, table S2). Consequently, inhibition or promotion of such protein expression could result in a broad spectrum of phenotypes. In addition, different geographical populations also need to adapt to a variety of other biotic and abiotic factors not considered in this study.

## 3.3. Population genetic analyses

The global average SNP diversity ($\pi$) of *S. marmoratus* was $0.136 \pm 0.064$ and ranged from 0.074 to 0.199 within each sample location. The global observed heterozygosity ($H_o$) and expected heterozygosity ($H_e$) were $0.130 \pm 0.077$ and $0.136 \pm 0.066$, respectively, and the estimates within each population ranged from 0.152 to 0.185 and from 0.158 to 0.202, respectively (figure 2). Genetic diversity estimates of Chinese populations were comparatively higher than those of Japanese populations. Among the 10 populations the highest diversity estimates were all detected in population ZS, even when 10 individuals per population were randomly subsampled (electronic supplementary material, table S3).

The pairwise $F_{st}$ values ranged from −0.0055 for the Yokosuka–Kochi comparison to 0.0319 for the Yokosuka–Zhoushan comparison, overall showing a low degree of genetic differentiation (table 2). However, all pairwise $F_{st}$ values were

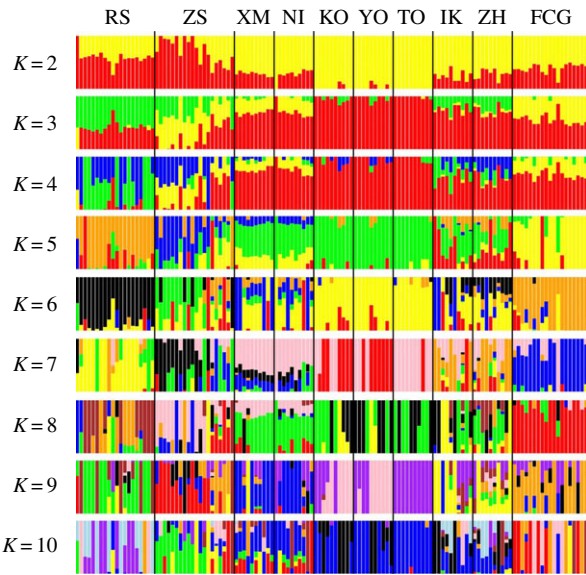

**Figure 3.** Population admixture analysis of 10 *S. marmoratus* populations based on the non-outlier dataset. Each bar represents an individual and each colour is inferred membership in each of the *K* (2–10) potential ancestral populations.

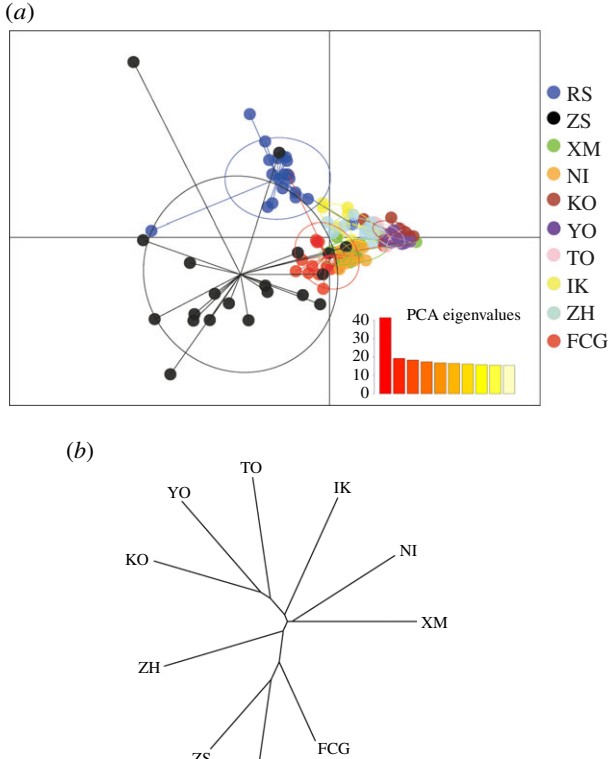

**Figure 4.** PCA plotting (*a*) and neighbour-joining topology (*b*) of 10 *S. marmoratus* populations based on the non-outlier dataset.

statistically significant after FDR correction ($p < 0.05$, FDR $\alpha = 0.01$) except for the Niigata–Xiamen and Yokosuka–Kochi pairs, indicating a certain degree of population differentiation.

Different methods of determining population structure generally produced similar results. The most likely numbers of the coancestry cluster were one (*K* = 1), suggesting population panmixia. As the *K* value increased, the plots showed six genetic subpopulations when *K* ranged from 5 to 10: (1) RS, (2) ZS, (3) XM, NI, (4) KO, YO and TO, (5) IK, ZH and (6) FCG (figure 3). The PCA recovered the same clusters achieved by Admixture analyses (figure 4*a*), showing genetic homogeneity among

sampled populations. The NJ tree mirrored the results of the Admixture and PCA analyses, showing shallow genetic structuring across all populations (figure 4*b*).

It is worth noting that the substructure of the Admixture analyses revealed an unexpectedly close relationship in two population pairs (i.e. XM-NI and ZH-IK), where the populations within the pairs have large geographical and latitudinal distances (figure 3). Given the relatively short geographical distance, a close genetic relationship should probably be detected in the XM-ZH and NI-IK pairs. The mismatch between genetic distance and geographical distance could be possible evidence for parallel adaptive evolution. Therefore, we further reconstructed outlier-based population structures of 10 populations and these four populations, respectively. PCA plotting revealed clear grouping within the XM-NI and ZH-IK pairs, which was consistent with clustering based on allele frequency of outliers (figure 5), showing evolutionary similarity of populations with large geographical and latitudinal gradients.

## 4. Discussion

In the present study, using high-resolution genomic SNPs, weak but statistically significant levels of genetic differentiation were detected among *S. marmoratus* populations. *S. marmoratus* is a demersal rockfish with restricted migration during the juvenile and adult life history and thus is considered to disperse only during the larval stage. However, given a very short pelagic larval duration lasting about 10 days [40], larval dispersal might also be restricted. Therefore, it is likely that the genetic similarity could be the result of limited divergence time from shared ancestry populations. The contemporary distribution ranges of *S. marmoratus* were almost exposed during the last glacial period [41], so this species may have been extirpated through large parts of its range and survived in glacial refugium. Population expansion and subdivision from glacial refugium are expected when favourable conditions returned and genetic homogeneity would also be expected in the recolonized regions given the relatively young postglacial ecosystems (less than 10 000 years) [41]. Under such a scenario, it can be inferred that the East China Sea glacial refugium, i.e. the Okinawa Trough, should be the centre of the diversity and origin of *S. marmoratus*, integrated with the highest genetic diversity of population ZS. Similar inferences were also derived in the population genetic studies of *Lateolabrax maculatus* [41], *Synechogobius ommaturus* [42] and *Thamnaconus hypargyreus* [43].

Understanding the capacity of natural populations to adapt to their local environment is a central topic in evolutionary biology [44]. Geographically distinct populations that are exposed to similar environmental conditions generally evolve similar genotypic and phenotypic traits [13]. In the present study, structuring analyses demonstrated genetic similarity in XM-NI and ZH-IK population pairs, in which populations span geographical distances of approximately 2000 km and latitudinal distances of approximately 10°. However, we failed to detect such genetic similarity in XM-ZH and NI-IK pairs. The close genetic relationship was also detected when using mitochondrial coding gene sequences [18]. Integrated with similar reproductive features and an outlier allele frequency pattern, the genetic similarity detected in XM-NI and ZH-IK population pairs could be considered possible evidence for parallel evolution in *S. marmoratus*. A similar conclusion was also drawn in the Atlantic herring case [13], in which genotypes

royalsocietypublishing.org/journal/rsob Open Biol. 9: 190028

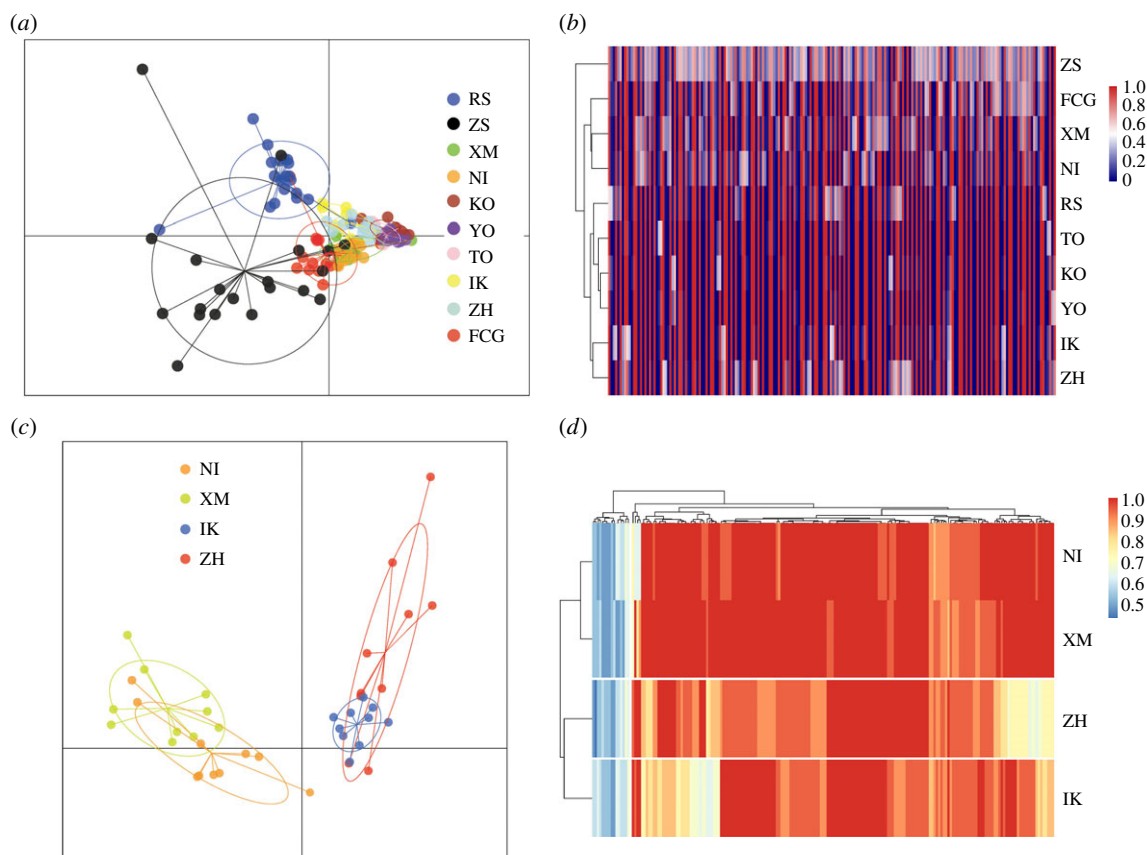

**Figure 5.** PCA plots and clustering of allele frequencies based on outliers. Hierarchical clustering was implemented in *heatmap* using Euclidean distance with the Ward clustering method. (*a*) PCA plotting in 10 populations; (*b*) allele frequency clustering in 10 populations; (*c*) PCA plotting in populations XM, ZH, NI and IK; (*d*) allele frequency clustering in populations XM, ZH, NI and IK.

were shared among geographically distant populations with low genetic differentiation. In addition, parallel genetic evolution was also reported in the threespine stickleback [8–10] and Atlantic cod [12], given similar environmental conditions (salinity in the threespine stickleback and temperature in Atlantic cod). Previous parallel evolution studies on marine organisms generally focused on populations in the same latitudinal gradients, ensuring that individuals undergo similar selective pressures. Herein, despite the high latitudinal interval (ca. 10°), possible parallel evolution evidence was also detected in *S. marmoratus* populations. To our knowledge, this might be the first report of parallel evolution in the NWP.

In spite of the large geographical and latitudinal interval, the habitats of SEC and JPN regions are generally homogeneous because of the influence of the KCS [14]. Environmental similarity might promote similar or identical genetic variants in the two regions [4]. The similar mating season [16,17] between the two regions could be an indicator of physiological and biological similarity. However, because of the low-efficiency annotation of the GBS technique, we failed to detect genetic variants associated with reproductive

biology in this study. Moreover, the complex and polygenic nature of parallel evolution restricted our understanding of genetic parallelism in *S. marmoratus*. Further studies linking environment, genotype and biology are warranted to fully demonstrate the pattern of parallel evolution in this species.

Data accessibility. The sequencing data have been deposited in the NCBI Sequence Read Archive (SRA) under accession nos. SRP095927, SRP110786 and SRP126707. The SNP matrices have been stored in Figshare (https://doi.org/10.6084/m9.figshare.8138219). The electronic supplementary material associated with this article is available in the online version.

Authors' contribution. T.G. and X.Z. designed the research. S.X., T.Y., S.C. and N.S. collected samples. S.X., S.C. and N.S. conducted analyses. S.X. and T.G. wrote the original manuscript. All authors contributed to revisions of the manuscript.

Competing interests. The authors declare no conflict of interest.

Funding. This study was supported by the National Natural Science Foundation of China (41776171; 41176117; 31172447).

Acknowledgements. We would like to thank Prof. Brian Bowen for valuable advice on the manuscript, and Drs Dongping Ji and Yuan Li for collecting samples.

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
