## [Reviewer comments · Open Biology]

Review History

RSOB-19-0028.R0 (Original submission)

Review form: Reviewer 1

Recommendation

Major revision is needed (please make suggestions in comments)

Are each of the following suitable for general readers?

- a) **Title**
Yes

- b) **Summary**
Yes

c) **Introduction**

Yes

Is the supplementary material necessary; and if so is it adequate and clear?

Yes

Do you have any ethical concerns with this paper?

No

Comments to the Author

Dear Editor and authors,

This manuscript entitled "Population genomics reveals possible genetic evidence for parallel evolution of *Sebastiscus marmoratus* in northwestern Pacific" analyzed population genetic structure and local adaptation of *Sebastiscus marmoratus* by genomic SNPs. The results are interesting, and can provide useful information for management and conservation of this species. I suggested acceptance after some revisions. There are some comments as follows.

1. Line 79, Illumina Hiseq2000 platform could output 100-bp paired-end reads. Please check and revise it.

2. Line 87-88 "SNP filtering was produced using VCFtools24 with the 88 following parameters: (1) the SNP was called in 50% of individuals (--max-missing 0.5)". The standard of 50% was too low, and please raise the standard.

3. For population genetic analyses, I would like to suggest the authors analyze the population structure of the 10 populations based the outlier loci, which could be compared with the population structure of the four populations.

4. Line 182-183, Isolation by distance theory don't suit many marine species, especially for low level genetic divergences among populations. Please rewrite the sentences "According to the isolation by distance theory, close genetic relationship should be detected in XM-ZH and NI-IK pairs. The mismatch between genetic distance and geographical distance could be possible evidence for parallel adaptive evolution."

5. Discussion should be written more about parallel adaptive evolution and listed more similar example.

6. Were there some similar morphological characteristics between the samples from XM-ZH and NI-IK populations?

Review form: Reviewer 2**Recommendation**

Major revision is needed (please make suggestions in comments)

Are each of the following suitable for general readers?a) **Title**

Yes

b) **Summary**

Yes

c) **Introduction**

Yes

Is the length of the paper justified?

Yes

Should the paper be seen by a specialist statistical reviewer?

No

Is it clear how to make all supporting data available?

No

Is the supplementary material necessary; and if so is it adequate and clear?

Yes

Do you have any ethical concerns with this paper?

No

Comments to the Author

Suggested revisions are attached (See Appendix A)

Decision letter (RSOB-19-0028.R0)

04-Mar-2019

Dear Dr Xu,

We are writing to inform you that the Editor has reached a decision on your manuscript RSOB-19-0028 entitled "Population genomics reveals possible genetic evidence for parallel evolution of *Sebastiscus marmoratus* in northwestern Pacific", submitted to Open Biology.

As you will see from the reviewers' comments below, there are a number of criticisms that prevent us from accepting your manuscript at this stage. The reviewers suggest, however, that a revised version could be acceptable, if you are able to address their concerns. If you think that you can deal satisfactorily with the reviewer's suggestions, we would be pleased to consider a revised manuscript.

The revision will be re-reviewed, where possible, by the original referees. As such, please submit the revised version of your manuscript within six weeks. If you do not think you will be able to meet this date please let us know immediately.

When submitting your revised manuscript, please respond to the comments made by the referee(s) and upload a file "Response to Referees" in "Section 6 - File Upload". You can use this to document any changes you make to the original manuscript. In order to expedite the

processing of the revised manuscript, please be as specific as possible in your response to the referee(s).

Please see our detailed instructions for revision requirements
<https://royalsociety.org/journals/authors/author-guidelines/>

Sincerely,

The Open Biology Team
mailto:openbiology@royalsociety.org

Reviewer(s)' Comments to Author(s):

Referee: 1

Comments to the Author(s)

Dear Editor and authors,

This manuscript entitled "Population genomics reveals possible genetic evidence for parallel evolution of *Sebastes marmoratus* in northwestern Pacific" analyzed population genetic structure and local adaptation of *Sebastes marmoratus* by genomic SNPs. The results are interesting, and can provide useful information for management and conservation of this species. I suggested acceptance after some revisions. There are some comments as follows.

1. Line 79, Illumina Hiseq2000 platform could output 100-bp paired-end reads. Please check and revise it.
2. Line 87-88 "SNP filtering was produced using VCFtools24 with the 88 following parameters: (1) the SNP was called in 50% of individuals (--max-missing 0.5)". The standard of 50% was too low, and please raise the standard.
3. For population genetic analyses, I would like to suggest the authors analyze the population structure of the 10 populations based the outlier loci, which could be compared with the population structure of the four populations.
4. Line 182-183, Isolation by distance theory don't suit many marine species, especially for low level genetic divergences among populations. Please rewrite the sentences "According to the isolation by distance theory, close genetic relationship should be detected in XM-ZH and NI-IK pairs. The mismatch between genetic distance and geographical distance could be possible evidence for parallel adaptive evolution."
5. Discussion should be written more about parallel adaptive evolution and listed more similar example.
6. Were there some similar morphological characteristics between the samples from XM-ZH and NI-IK populations?

Referee: 2

Comments to the Author(s)

Suggested revisions are attached

Author's Response to Decision Letter for (RSOB-19-0028.R0)

See Appendix B.

RSOB-19-0028.R1 (Revision)

Review form: Reviewer 1

Recommendation

Accept with minor revision (please list in comments)

Are each of the following suitable for general readers?

- a) **Title**
Yes
- b) **Summary**
Yes
- c) **Introduction**
Yes

Is the length of the paper justified?

Yes

Should the paper be seen by a specialist statistical reviewer?

Yes

Is it clear how to make all supporting data available?

Yes

Is the supplementary material necessary; and if so is it adequate and clear?

Yes

Do you have any ethical concerns with this paper?

No

Comments to the Author

This manuscript entitled "Population genomics reveals possible genetic evidence for parallel evolution of *Sebastiscus marmoratus* in northwestern Pacific" analyzed population genetic diversity, structuring and local adaptation of *Sebastiscus marmoratus* by investigating genomewide SNPs. The results showed possible evidence for parallel genetic evolution of this species, which are interesting, and can provide novel and useful information for management of this species. I suggested acceptance after some minor revisions, as follows:

1. Line 87-92, the parameter scripts can be provided in supplementary information.
2. Line 159, the proportion of annotated loci is very low, the authors also said in line 238 "due to the low-efficiency annotation of GBS technique, genetic variants associated with reproductive biology were failed to detect in this study". What is the main reason according to the authors'

opinion?

3. Were there some similar morphological or biological characteristics between the samples from the two population pairs XM-ZH and NI-IK? Integrated phenotypic and genetic evidence, the parallel evolution of this species can be more persuasive.

4. Line 232, the authors stated that "this might be the first report for parallel evolution in the Northwestern Pacific", is there any information of possible parallelism in other marine species? If so, the Northwestern Pacific could be a good region for convergence studies.

5. The format of "northwestern Pacific" should be unified in the manuscript, such as "Northwestern Pacific" or "the Northwestern Pacific".

Decision letter (RSOB-19-0028.R1)

23-Jul-2019

Dear Dr Xu

We are pleased to inform you that your manuscript RSOB-19-0028.R1 entitled "Population genomics reveals possible genetic evidence for parallel evolution of *Sebastiscus marmoratus* in northwestern Pacific" has been accepted by the Editor for publication in Open Biology.

The referee suggests minor revisions at outlined below. We would also like to request that the authors carry out a final proof-read of the manuscript and upload the final files for publication.

Please submit the revised version of your manuscript within 7 days. If you do not think you will be able to meet this date please let us know immediately and we can extend this deadline for you.

Please see our detailed instructions for revision requirements. It is essential these instructions are followed carefully to minimise any delay to publication:
<http://rsob.royalsocietypublishing.org/site/misc/revised.xhtml>

- 1) A text file of the manuscript (doc, txt, rtf or tex), including the references, tables (including captions) and figure captions. Please remove any tracked changes from the text before submission. PDF files are not an accepted format for the "Main Document".
- 2) A separate electronic file of each figure (tiff, EPS or print-quality PDF preferred). The format should be produced directly from original creation package, or original software format. Please note that PowerPoint files are not accepted.
- 3) Electronic supplementary material: this should be contained in a separate file from the main text and meet our ESM criteria (see

<http://rsob.royalsocietypublishing.org/site/misc/styleandpolicy.xhtml#question11>). Please note that ESM files are NOT edited by the Royal Society so should be submitted as the authors intend readers to view them with accompanying title/s and caption/s. Where possible we request that authors combine multiple ESM files into one file (for example, where ESM files are in Word or PDF format). In addition, the number of references included in the ESM should be kept to an absolute minimum as these are not recognised by many indexing services.

4) A media summary: a short non-technical summary (up to 100 words) of the key findings/importance of your manuscript. Please try to write in simple English, avoid jargon, explain the importance of the topic, outline the main implications and describe why this topic is newsworthy.

Images

We require suitable relevant images to appear alongside published articles. Do you have an image we could use? Images should be approximately 200 mm x 300 mm, be in a four-colour format and have a resolution of at least 300 dpi.

Sincerely,

The Open Biology Team
<mailto:openbiology@royalsociety.org>

Reviewer's Comments to Author:

Referee:

Comments to the Author(s)

This manuscript entitled "Population genomics reveals possible genetic evidence for parallel evolution of *Sebastiscus marmoratus* in northwestern Pacific" analyzed population genetic diversity, structuring and local adaptation of *Sebastiscus marmoratus* by investigating genomewide SNPs. The results showed possible evidence for parallel genetic evolution of this species, which are interesting, and can provide novel and useful information for management of this species. I suggested acceptance after some minor revisions, as follows:

1. Line 87-92, the parameter scripts can be provided in supplementary information.
2. Line 159, the proportion of annotated loci is very low, the authors also said in line 238 "due to the low-efficiency annotation of GBS technique, genetic variants associated with reproductive biology were failed to detect in this study". What is the main reason according to the authors' opinion?
3. Were there some similar morphological or biological characteristics between the samples from the two population pairs XM-ZH and NI-IK? Integrated phenotypic and genetic evidence, the parallel evolution of this species can be more persuasive.
4. Line 232, the authors stated that "this might be the first report for parallel evolution in the Northwestern Pacific", is there any information of possible parallelism in other marine species? If so, the Northwestern Pacific could be a good region for convergence studies.
5. The format of "northwestern Pacific" should be unified in the manuscript, such as "Northwestern Pacific" or "the Northwestern Pacific".

Author's Response to Decision Letter for (RSOB-19-0028.R1)

See Appendix C.

Decision letter (RSOB-19-0028.R2)

12-Aug-2019

Dear Dr Xu

We are pleased to inform you that your manuscript entitled "Population genomics reveals possible genetic evidence for parallel evolution of *Sebastes marmoratus* in northwestern Pacific" has been accepted by the Editor for publication in Open Biology.

Article processing charge

Please note that the article processing charge is immediately payable. A separate email will be sent out shortly to confirm the charge due. The preferred payment method is by credit card; however, other payment options are available.

Sincerely,

The Open Biology Team
mailto: openbiology@royalsociety.org

Appendix A

This is an evolutionary study of *Sebasticus marmoratus*, a fish in the NW Pacific Ocean where a cryptic lineage is uncovered.

The sample sizes aren't high, but this did not seem to preclude the detection of a signal. However, there are methodological issues that need to be addressed in the filtering of data for statistical analysis. The biggest issue is that linkage disequilibrium is not accounted for in any manner, and thus the assumptions of the FST and Admixture analyses are being violated. Willis et al 2018 (Haplotyping RAD loci: an efficient method to filter paralogs and account for physical linkage) explicitly show that admixture analyses are sensitive to this. The remedy is to analyze 1 SNP per contig or linkage group, or analyze haplotype data. When the analyses are rerun, then it will be possible to evaluate the results and the conclusions based upon them.

I'd also like to point out that it is important that a study be reproducible, and I encourage the authors to provide their data processing/filtering scripts as supplements and fully report on their computation and laboratory methods. It is impossible to evaluate the validity of the methods employed as the manuscript stands.

Abstract:

The English grammar would benefit from a fair bit of editing.

20: be more specific, what is meant by "high minor allelic frequency"?

Methods:

72-80: more details are necessary here. What were the digest conditions? How were the barcodes attached? How long were the barcodes? What were the adapter sequences? Same for PCR. Conditions? Primer sequences? Number of cycles?

81-92: More details are necessary. Each of the programs used to process the data have myriad settings which need to be reported, otherwise this work cannot be reproduced. The script used to process the data should be provided as supplementary material or be deposited in a public database.

81-92: There seems to have been no attempt to account for the linkage between SNPs which is important for some analyses, such as global FST calculation, Admixture, and others. This needs to be addressed and is customarily handled by either selecting 1 SNP per contig. Given that a draft genome is available, the authors should be able to go a bit farther by assessing the linkage among contigs based on proximity or using linkage disequilibrium.

108-109: Nucleotide diversity can only be calculated with knowledge of the sequence length, not only the number of segregating sites. Thus, while you may have asked arlequin to calculate nucleotide diversity, it does not know the sequence length if you only gave it SNP data, and thus the values calculated by arlequin are not nucleotide diversity. This needs to be rectified.

110: could rarefaction be used here instead?

113-115: specify what the clustering on allele frequencies is intended to accomplish

148-149: while you are doing what everybody else does, I'd like to point out that the outlier detection methods used are framed to make it very difficult for a locus to be considered an outlier. Thus, there are many loci that are responding to selection that are retained in the data set. At the very least, rather than "neutral" you could use "non-outlier" as a descriptor that would be much more accurate.

Results:

160: as noted above, nucleotide diversity was not calculated in this study. It either needs to be called something else (mean SNP diversity) or the lengths of the sequences need to be accounted for.

Appendix B

List of Responses

Dear Editors and Reviewers:

Thank you very much for your letter and for the reviewers' comments concerning our manuscript entitled "**Population genomics reveals possible genetic evidence for parallel evolution of *Sebastiscus marmoratus* in northwestern Pacific**" (ID: RSOB-19-0028). Those comments are all valuable and very helpful for revising and improving our paper, as well as the important guiding significance to our researches. We have studied comments carefully and have made correction which we hope meet the approval. Revised portion are marked in red in the paper. The main corrections in the paper and the responses to the reviewer's comments are as following:

Responds to the reviewer's comments:

Reviewer #1:

1. Response to comment: Line 79, Illumina Hiseq2000 platform could output 100-bp paired-end reads. Please check and revise it.

Response: We are very sorry for that. According to the reviewer's suggestion, we have reworded the "Hiseq2000 platform" to "Hiseq2500 platform".

2. Response to comment: Line 87-88 " SNP filtering was produced using VCFtools24 with the 88 following parameters: (1) the SNP was called in 50% of individuals (--max-missing 0.5)". The standard of 50% was too low, and please raise the standard.

Response: We are very thankful for the reviewer's comments. We also referred to some population genomics literatures and have changed the standard to 90%. But the number of SNPs were the same. We have provided the filtering scripts in the revised version to make our results reproducible, as per Reviewer #2's suggestion.

3. Response to comment: For population genetic analyses, I would like to suggest the authors analyze the population structure of the 10 populations based the outlier loci, which could be compared with the population structure of the four populations.

Response: According to the reviewer's suggestion, we have supplemented the population structure (PCA plotting and Heatmap clustering) of 10 populations based on outlier loci.

4. Response to comment: Line 182-183, Isolation by distance theory don't suit many marine species, especially for low level genetic divergences among populations. Please rewrite the sentences "According to the isolation by distance theory, close genetic relationship should be detected in XM-ZH and NI-IK pairs. The mismatch between genetic distance and geographical distance could be possible evidence for parallel adaptive evolution."

Response: According to the reviewer's suggestion, we have revised this sentence.

5. Response to comment: Discussion should be written more about parallel adaptive evolution and listed more similar example.

Response: According to the reviewer's suggestion, we have revised the discussion to add more parallel evolution examples.

6. Response to comment: were there some similar morphological characteristics between the samples from XM-ZH and NI-IK populations?

Response: We didn't check the morphological characteristics between the samples from XM-ZH and NI-IK population pairs, since we only had the tissue samples of Japanese individuals. Further studies are warranted to focus on specific genes associated with biological features such as reproductive biology.

Special thanks to you for your good comments!

Reviewer #2:

1. Response to comment: The samples sizes aren't high, but this did not seem to preclude the detection of a signal. However, there are methodological issues that need to be addressed in the filtering of data for statistical analysis. The biggest issue is that linkage disequilibrium is not accounted for in any manner, and thus the assumptions of the FST and Admixture analyses are being violated. Willis et al 2018 (Haplotyping RAD loci: an efficient method to filter paralogs and account for physical linkage) explicitly show that admixture analyses are sensitive to this. The remedy is to analyze 1 SNP per contig or linkage group, or analyze haplotype data. When the analyses are rerun, then it will be possible to evaluate the results and the conclusions based upon them.

Response: Thank you very much for the suggestion. According the reviewer's suggestion, we have analyzed linkage disequilibrium by using TASSEL software in our revised version. As a result, a total of 574 SNPs exhibited strong LD and were excluded in further analyses.

2. Response to comment: I'd also like to point out that it is important that a study be reproducible, and I encourage the authors to provide their data processing/filtering scripts as supplements and fully report on their computation and laboratory methods. It is impossible to evaluate the validity of the methods employed as the manuscript stands.

Response: We are very thankful for the reviewer's comment. However, due to the Illumina library preparation and sequencing were preformed commercially in Novogene Co. Ltd in Beijing, some of the steps were commercially confidential data. Therefore, we failed to supplement the details and data preprocessing scripts in the revised version. But as per your comment, we provide the SNP filtering scripts in the revised manuscript.

3. Response to comment: The English grammar would benefit from a fair bit of editing.

Response: According to the reviewer's suggestion, we have modified the Abstract part.

4. Response to comment: 20: be more specific, what is meant by "high minor allelic frequency"?

Response: We are very thankful for the reviewer's comment. Considering that biallelic SNPs with rare minor alleles are expected to yield small F_{st} values, we firstly set the threshold of MAF to 10% (high minor allelic frequency). However, we referred to some population genomics literatures and set the threshold to 5% in the revised version.

5. Response to comment: 72-80: more details are necessary here. What were the digest conditions? How were the barcodes attached? How long were the barcodes? What were the adapter sequences? Same for PCR. Conditions? Primer sequences? Number of cycles?

Response: We are very thankful for the reviewer's comment. Just as mentioned in Response 2, the Illumina library preparation and sequencing were preformed commercially in Novogene Co. Ltd in Beijing, and some of the steps were commercially confidential data. Therefore, we failed to supplement the details and data preprocessing scripts in the revised version. We are sorry for that.

6. Response to comment: 81-92: More details are necessary. Each of the programs used to process the data have myriad settings which need to be reported, otherwise this work cannot be reproduced. The script used to process the data should be provided as supplementary material or be deposited in a public database.

Response: According to the reviewer's suggestion, we have provided the data analyses scripts in the revised manuscript.

7. Response to comment: 81-92: There seems to have been no attempt to account for the linkage

between SNPs which is important for some analyses, such as global F_{ST} calculation, Admixture, and others. This needs to be addressed and is customarily handled by either selecting 1 SNP per contig. Given that a draft genome is available, the authors should be able to go a bit farther by assessing the linkage among contigs based on proximity or using linkage disequilibrium.

Response: According to the reviewer's suggestion, we have analyzed linkage disequilibrium by using TASSEL software in our revised version and a total of 574 SNPs exhibited strong LD and were excluded in population genomics analyses.

8. Response to comment: 108-109: Nucleotide diversity can only be calculated with knowledge of the sequence length, not only the number of segregating sites. Thus, while you may have asked arlequin to calculate nucleotide diversity, it does not know the sequence length if you only gave it SNP data, and thus the values calculated by arlequin are not nucleotide diversity. This needs to be rectified; 160: as noted above, nucleotide diversity was not calculated in this study. It either needs to be called something else (mean SNP diversity) or the lengths of the sequences need to be accounted for.

Response: According to the reviewer's suggestion, we have reworded "nucleotide diversity" to "average SNP diversity" in the revised version.

9. Response to comment: 110: could rarefaction be used here instead?

Response: We are very thankful for the reviewer's comment. According to the reviewer's comment, we have rewritten this sentence in the revised manuscript.

10. Response to comment: 113-115: specify what the clustering on allele frequencies is intended to accomplish.

Response: According to the reviewer's suggestion, we have rewritten this sentence in the revised manuscript.

11. Response to comment: 148-149: while you are doing what everybody else does, I'd like to point out that the outlier detection methods used are framed to make it very difficult for a locus to be considered an outlier. Thus, there are many loci that are responding to selection that are retained in the data set. At the very least, rather than "neutral" you could use "non-outlier" as a descriptor that would be much more accurate.

Response: Thank you very much for the reviewer's comment. According to the reviewer's suggestion, we have reworded "neutral" to "non-outlier" in the revised version.

Special thanks to you for your good comments!

We appreciate for editors and reviewers' warm work earnestly, and hope that the correction will meet with approval.

Once again, thank you very much for your comments and suggestions.

Appendix C

List of Responses

Dear Editors and Reviewers:

Thank you very much for your letter and for the reviewers' comments concerning our manuscript entitled "**Population genomics reveals possible genetic evidence for parallel evolution of *Sebastiscus marmoratus* in northwestern Pacific**" (ID: RSOB-19-0028.R1). We have studied comments carefully and have made correction which we hope meet the approval. Revised portion are marked in red in the paper. The main corrections in the paper and the responses to the reviewer's comments are as following:

Responds to the reviewer's comments:

Reviewer #1:

1. Response to comment: Line 87-92, the parameter scripts can be provided in supplementary information.

Response: According to the reviewer's suggestion, the parameters scripts were shown in supplementary information (also see Line 92-93 in the revised manuscript).

2. Response to comment: Line 159, the proportion of annotated loci is very low, the authors also said in line 238 "due to the low-efficiency annotation of GBS technique, genetic variants associated with reproductive biology were failed to detect in this study". What is the main reason according to the authors' opinion?

Response: We are very thankful for the reviewer's comments. In our opinion, the low proportion of annotated loci might be due to the combination of lack of high-quality reference genome and short assembled contigs of GBS reads (also see Line 158-159 in the revised manuscript).

3. Response to comment: Were there some similar morphological or biological characteristics between the samples from the two population pairs XM-ZH and NI-IK? Integrated phenotypic and genetic evidence, the parallel evolution of this species can be more persuasive.

Response: We didn't check the morphological characteristics between the samples from XM-ZH and NI-IK population pairs, since we only had the tissue samples of Japanese individuals. Further studies are warranted to focus on specific genes associated with biological features such as reproductive biology.

4. Response to comment: Line 232, the authors stated that "this might be the first report for parallel evolution in the Northwestern Pacific", is there any information of possible parallelism in other marine species? If so, the Northwestern Pacific could be a good region for convergence studies.

Response: We are very thankful for the reviewer's comments. To our knowledge, numbers of warm-temperate marine fish species (such as wrasses) distributed in Japanese coastal waters and Chinese Taiwan-Fujian coastal waters, yet no record were reported in the north part of East China Sea. This might be also resulted from the influence of Kuroshio Current system. We argue that marine organisms in northwestern Pacific should be good materials for parallel evolution.

5. Response to comment: The format of "northwestern Pacific" should be unified in the manuscript, such as "Northwestern Pacific" or "the Northwestern Pacific".

Response: According to the reviewer's suggestion, we have revised the manuscript.

Special thanks to you for your good comments!

We appreciate for editors and reviewers' warm work earnestly, and hope that the correction will meet with approval.

Once again, thank you very much for your comments and suggestions.